# (GIGA)byte

DATA RELEASE

# Jellyfish in Hong Kong: a citizen science dataset

John Terenzini[1,*,†], Yannan Fan[2], Melissa Jean-Yi Liu[3] and Laura J. Falkenberg[4,5]

1 Hong Kong Jellyfish Project, Hong Kong, China
2 GigaScience Press, BGI Center, Meisha Street, Yantian district, Shenzhen, Guangdong, 518000, China
3 Taiwan Biodiversity Information Facility (TaiBIF), Biodiversity Research Center, Academia Sinica, 128 Sec. 2, Academia Rd., Nangang, Taipei, 115, Taiwan
4 Simon FS Li Marine Science Laboratory, School of Life Sciences, Chinese University of Hong Kong, Hong Kong, China
5 UniSA STEM, University of South Australia, Australia

## ABSTRACT

The Hong Kong Jellyfish Project is a citizen science initiative started in early 2021 to enhance our understanding of jellyfish in Hong Kong. Here, we present a dataset of jellyfish sightings collected by citizen scientists from 2021 through 2023 within local waters. Citizen scientists submitted photographs and other data (time, date, and location) using a website, iNaturalist project, and social media. Sightings were validated using references from the literature. A total of 1,020 usable observations are included in this dataset, showing the occurrence and distribution of jellyfish in Hong Kong in 2021–2023. This dataset is now publicly available and discoverable in the Global Biodiversity Information Facility database and is available for download. This data can be used to enhance our understanding of the biodiversity of local marine ecosystems.

**Subjects** Animal and Plant Sciences, Biodiversity, Marine biology

**Submitted:** 19 February 2024

\* Corresponding author. E-mail: project@hkjellyfish.com

† Current address: State Key Laboratory of Marine Pollution, The City University of Hong Kong, Hong Kong, China.

Preprint submitted at https://doi.org/10.5281/zenodo.11202825

## DATA DESCRIPTION

### Background

Collecting biodiversity data is necessary for a clear understanding of species presence, distribution, and threats, allowing for informed conservation decisions at both a local and global scale. The Global Biodiversity Information Facility (GBIF) is the largest biodiversity database in the world, providing free and open access to biodiversity data for research and policy analyses. GBIF occurrence records originate from a variety of sources, such as specimen collection and literature records. Recently, citizen science has become GBIF's biggest source of data [1]. The biggest sources of citizen science-derived databases in GBIF are eBird and iNaturalist, which provide large volumes of data to GBIF while also introducing biases in data types (e.g., birds are emphasized) [1]. There are also some smaller and mid-sized citizen science projects sharing their biodiversity data in GBIF (e.g., the TurtleSpot Taiwan [2] and the Taiwan Roadkill Observation Network [3]). In addition to these projects, more are needed to fill persisting spatial and taxonomic biodiversity gaps.

One such effort is the Hong Kong Jellyfish Project (HKJP), which launched in early 2021, allowing citizen scientists to record the presence, abundance, and distribution of jellyfish in Hong Kong (for an introduction, see the video linked in Figure 1). The uploaded dataset contains observations made by citizen scientists between 2021 and 2023, comprising 1,020

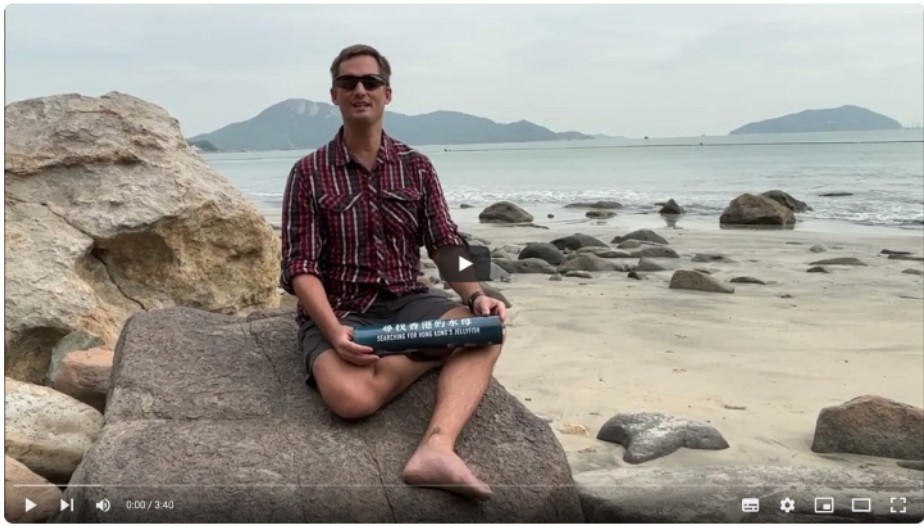

John Terenzini on the Hong Kong Jellyfish Project

**Figure 1.** Video introduction about the Hong Kong Jellyfish Project [10]. https://youtu.be/-4gH5fnO7WI

usable observations. These observations were recorded through the HKJP website [4], collected by the iNaturalist project [5], and gathered from social media. Observations that did not contain verifiable photographs, non-medusa taxa, or were taken from captivity were eliminated from the dataset (see Data validation and quality control). Ensuring accurate photographs is necessary for verification of identification. When these photographs are accessible in online databases, they can be used to test hypotheses in biology, ecology, and niche modeling [6]. Usable observations were identified to the lowest taxonomic level and verified by the HKJP principal investigator with reference to the literature primarily [6–9]. See Taxonomic scope for the results. It is important to have a dataset curated by specialists because, while citizen scientists can provide valuable observations, quality control is necessary to ensure accurate identifications and to validate the dataset (See Data validation and quality control), especially for uncommon taxa such as jellyfish.

This Data Release describes a dataset bringing together observations submitted through the HKJP website, the iNaturalist project, and gathered from social media. These observations were then harmonized and deposited in the GBIF repository.

## CONTEXT

Jellyfish are being studied in greater detail worldwide as their ecological and societal importance becomes better understood [11]. Due to the difficulties in finding jellyfish in the vast ocean and predicting their occurrence, the involvement of citizen scientists can be an effective way of spotting jellyfish. Citizen science data can be used to inform new species area records, as well as to understand seasonal and geographical distribution patterns of jellyfish [12]. Some professional scientists have critical views of citizen science because of concerns about data quality, biases in the available data, or a lack of interest in engaging with the public to conduct research [13]. However, citizen scientists have been found to have datasets of similar quality to professional scientists when provided with a suitable framework and robust data verification practices [14]. Members of the public are

sometimes unsure about engaging with scientific research because of the amount of time required and a lack of knowledge [15]; however, by providing variable levels of participation, educational opportunities, and an understanding of the benefits of the project, citizen scientists can be encouraged to greater levels of participation [16]. Overall, with suitable best practices in place when designing and executing citizen science projects, the benefits of engaging citizen scientists in marine research far outweigh the potential challenges [17].

Recording jellyfish seasonality and distribution can better inform our understanding of marine ecosystems. Notably, changes to their seasonality and distribution can be indicative of changes in ocean environments [18] and have major impacts on other marine species, including some of economic importance [19]. In Hong Kong during the 1980s, jellyfish proliferation and intake to the China Light & Power's Castle Peak Station posed a significant issue, prompting research into the seasonality of jellyfish in the area [20]. Large blooms of jellyfish are also known to have short- and long-term impacts on tourism [21], as well as fisheries and aquaculture [22].

Jellyfish records in Hong Kong are largely based on reports from the 1980s and 1990s [23, 24], which are documented in the Hong Kong Register of Marine Species [25]. Since then, our understanding of local jellyfish comes from incidental individual encounters [26] and lab-based research on their genes [27]. The dataset reported here represents an initiative based on citizen science observations and is shared to enable a better understanding of jellyfish presence, seasonality, and distribution in Hong Kong. Citizen science is known to have a strong bias toward vertebrates and terrestrial ecosystems [28]; hence, collecting data of marine invertebrates such as jellyfish should be particularly useful in helping fill these missing taxonomic data gaps in the global biodiversity databases [29].

## METHODS

### Study area

Hong Kong Special Administrative Region is located on the southern coast of mainland China (22.3193°N, 114.1694°E) at the northern end of the South China Sea. Its marine boundary contains 1,651 km$^2$ of water [25] and a rich marine biodiversity despite constituting only 0.03% of Chinese waters [30]. The location of Hong Kong in the South China Sea means there is an overlap of species associated with tropical and temperate climates [31]. Due to Hong Kong's status as a major maritime port with highly disturbed shorelines, there is also a high potential for invasive marine species to be present [32]. The biodiversity of Hong Kong is extensively studied; however, gaps remain in the publicly available records, particularly in terms of species records and how they are changing over time, with the potential for previously unrecorded species to be found [33]. All jellyfish observations from the HKJP were made in Hong Kong waters.

### Sampling

Participants submit any observation of an individual jellyfish, a group of jellyfish, or multiple observations of different jellyfish at their own discretion (Figure 2). Additionally, observations of the absence of jellyfish can be reported through the HKJP website or social media, and these are captured in GBIF. Jellyfish are encountered opportunistically by participants as they conduct their daily activities. Observations typically increase during

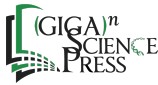

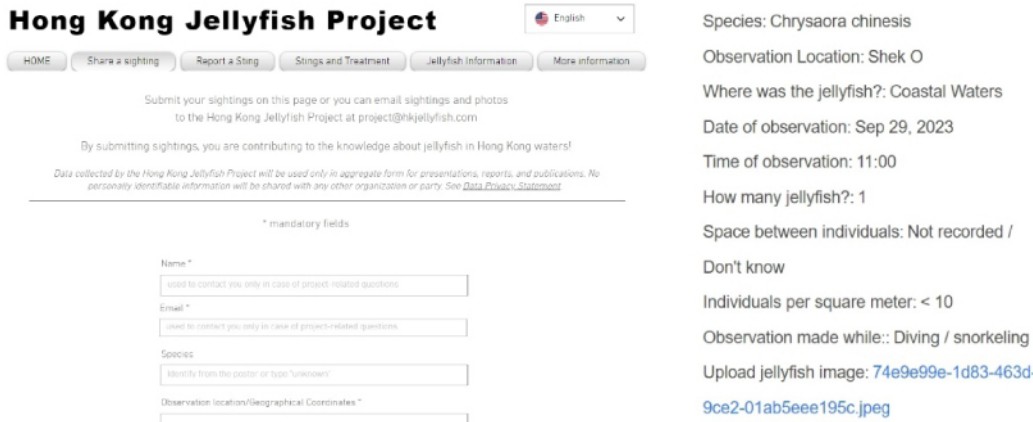

**Figure 2.** Observation submission page of the Hong Kong Jellyfish Project (https://www.hkjellyfish.com/share-a-sighting) and an example observation.

high bloom times in April and May, following a seasonal pattern [12]. Sightings are also reported following the project promotion through social (Facebook, Instagram) and traditional media (radio, newspaper). Periodic HKJP newsletters by email and journal posts on iNaturalist are used to maintain observer interest and promote project awareness.

Each observation from the HKJP website includes fields describing their: (i) taxonomy to the lowest level possible (kingdom, phylum, class, order, family, genus, species); (ii) location in observer's own words or geolocation (latitude, longitude); (iii) date of observation (dd/mm/yyyy); (iv) time of observation (24 hr); (v) how many jellyfish were seen (no jellyfish seen, 1, <10, 10–100, 100–500, 500–1000, not recorded/don't know); (vi) space between individuals (10 cm, <1 m, 1–5 m, 5–10 m, 10–20 m, >20 m, not recorded/don't know); (vii) individuals per square meter (<10, 10–100, 100–500, <500, not recorded/don't know); (viii) observation made while (fishing, sailing, diving/snorkeling/swimming, on a boat/ferry, walking along the coast, kayaking/Stand-Up Paddleboarding, other); (ix) photograph(s) and/or video of jellyfish. For observations that came through social media, we attempted to obtain as much of the above information as possible, though greater gaps remain in this dataset.

Observations from the iNaturalist project include fields describing: (i) user id; (ii) date of observation (dd/mm/yyyy); (iii) time of observation (24 hr); (iv) URL of observation; (v) image URL; (vi) place guess; (vii) latitude of the observation; (viii) longitude of the observation; (ix) species guess; (x) the scientific name to the lowest level possible (kingdom, phylum, class, order, family, genus, species); (xi) the common name as determined by iNaturalist.

The observation data has been compiled into the Darwin Core format with fields including: (i) occurrenceID; (ii) basisOfRecord; (iii) type of observation; (iv) language; (v) institutionID; (vi) institutionCode; (vii) recordedBy; (viii) eventDate; (ix) eventTime; (x) associatedReferences; (xi) associatedMedia; (xii) occurrenceRemarks; (xiii) locality; (xiv) decimalLatitude; (xv) decimalLongitude; (xvi) geodeticDatum; (xvii) country; (xviii) stateProvince; (xix) vernacularName; (xx) scientificName; (xxi) acceptedNameUsageID; (xxii) taxonRank; (xxiii) kingdom. Explanations for each Darwin Core field can be found on their website [34].

### Step description

(1)  A citizen scientist observes jellyfish (or the absence of jellyfish) while undertaking an activity such as swimming, kayaking, or walking along the shore.

(2)  The observer takes a photograph of the jellyfish and uploads it either through the HKJP website, the iNaturalist app on their phone, or to social media (Facebook [35] and Instagram - @hkjellyfishproject). If recording through the website or iNaturalist app, observers note the date, time, location, and species, if known. Additional information, including the number and density of jellyfish, may be optionally provided if a large number of them is present.

(3)  A submission reported through the website is reviewed by the HKJP principal investigator for identification verification, and, if necessary, reference is made to the literature. See Data validation and quality control for a description of the process. Confirmation and additional species information are then shared with the observer. If an observation cannot be identified, it is included in the dataset, but the lack of identification is clearly indicated.

(4)  For a submission reported through iNaturalist, artificial intelligence usually suggests an identification or a category after automatically recording location and time data within the observer's pre-set parameters. The observer can select the suggested identification or make their own. Anyone on iNaturalist can suggest an identification, and, when multiple people agree, the identification is considered Research Grade. However, since agreement among iNaturalist users does not necessarily provide accurate identification, all identifications from iNaturalist are reviewed by the HKJP principal investigator before entering them into the dataset.

(5)  For a submission reported on social media, the HKJP principal investigator asks for as much information as possible about the observed jellyfish (such as time, date, and location) and permission for use.

(6)  Observation data from all three sources is recorded into an Excel spreadsheet in English. A total of 1,020 usable observations have been recorded. Website and iNaturalist submissions can be entered in Traditional Chinese; hence, such observations are translated to English, the language of the HKJP's principal investigator, for inclusion in the spreadsheet.

(7)  The various data sources are harmonized and curated together into one table, which is then converted into Darwin Core Archive standard to be uploaded to GBIF.

### DATA VALIDATION AND QUALITY CONTROL

Observations that did not contain verifiable photographs, non-medusa taxa, or were taken from captivity were eliminated from the dataset. For example, some photographs were out-of-focus, making identification impossible, or of non-jellyfish taxa that had been misidentified by iNaturalist's artificial intelligence and not updated by the user. Since these could not be identified or were not jellyfish, they were removed from the dataset. Additionally, some photographs were taken at the jellyfish exhibit at a local theme park, Ocean Park, and uploaded to iNaturalist. Since these jellyfish were not in a natural ecosystem, the observations were removed from the dataset. The entered coordinates of five observations were erroneously placed outside the study area; hence, four of these observations were updated to correspond with the Locality information, and one observation without Locality information was removed. Usable observations were

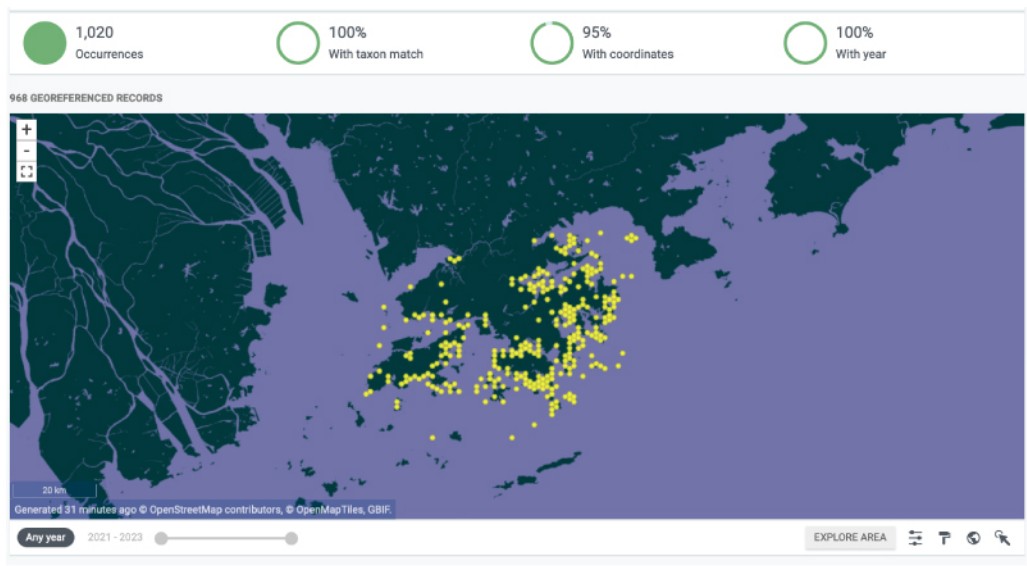

**Figure 3.** Interactive map of the georeferenced occurrences hosted by GBIF [36] https://www.gbif.org/dataset/e85c3b78-b780-4955-a50b-4adf2830a07c.

identified to the lowest taxonomic level and verified with reference to the literature (primarily [7–9] and other sources as required). Some observations were verified by sampling and DNA analysis done at the Simon F.S. Li Laboratory at The Chinese University of Hong Kong [12].

The dataset is available in Darwin Core format; 1,020 terms are available for the 2021–2023 dataset. All mandatory fields are present and have undergone validation and screening using the TaiBIF Integrated Publishing Toolkit (IPT) before uploading and publishing via the GigaScience Press GBIF page [36] (Figure 3).

## Taxonomic scope
The HKJP records the presence of 46 jellyfish in Hong Kong. Specifically, 17 scyphozoans (two to genus level), 14 hydrozoans (one to genus level), three cubozoans (one to genus level), seven ctenophores (three to genus level), and five siphonophores (1 to genus level) (Table 1).

## RE-USE POTENTIAL
These data are important to address the research gap in jellyfish presence and distribution in Hong Kong by updating old records and adding new ones. This data can be downloaded from GBIF [36] and used by academic researchers, government groups, and watersports groups to better understand local jellyfish presence and distribution. This dataset adds to the known jellyfish in Hong Kong for a better understanding of marine ecosystem composition with potential implications for fisheries management [37]. The temporal data in the dataset supports research into the seasonality of local jellyfish [12], which has been shown to have effects on the local industry [20]. Combining seasonal and geographic data with environmental data could be used to predict jellyfish occurrence and inform recreational users of marine environments [38]. Because of the difficulties in engaging in marine science due to potentially high costs and the need for specialized equipment, as well



**Table 1.** Taxonomic scope of jellyfish identified to genus and species level for each class identified (in bold) for the Hong Kong Jellyfish Project.

| Order | Family | Genus | Species |
|---|---|---|---|
| **Scyphozoa** | | | |
| Rhizostomeae | Catostylidae | *Acromitus* | *Acromitus flagellatus* |
| | Rhizostomatidae | *Rhopilema* | *Rhopilema hispidum* |
| | Lobonemidae | *Lobonema* | *Lobonema smithii* |
| | Lychnorhizidae | *Anomalorhiza* | *Anomalorhiza shawi* |
| | Cepheidae | *Cephea* | *Cephea cephea* |
| | | *Netrostoma* | *Netrostoma setouchianum* |
| | Cassiopeidae | *Cassiopeia* | |
| | Leptobrachidae | *Thysanostoma* | *Thysanostoma loriferum* |
| | Mastigiidae | *Mastigias* | *Mastigias papua* |
| | | *Phyllorhiza* | *Phyllorhiza punctata* |
| Semaeostomeae | Cyaneidae | *Cyanea* | *Cyanea nozakii* |
| | | *Cyanea* | *Cyanea purpurea* |
| | Pelagiidae | *Pelagia* | *Pelagia noctiluca* |
| | | *Chrysaora* | *Chrysaora chinensis* |
| | Ulmaridae | *Aurelia* | |
| | | *Diplulmaris* | *Diplulmaris malayensis* |
| Coronatae | Nausithoidae | *Nausithoe* | *Nausithoe punctata* |
| **Hydrozoa** | | | |
| Limnomedusae | Geryoniidae | *Liriope* | *Liriope tetraphylla* |
| | | *Olindias* | |
| Narcomedusae | Solmundaeginidae | *Solmundella* | *Solmundella bitentaculata* |
| Trachymedusae | Rhopalonematidae | *Aglaura* | *Aglaura hemistoma* |
| | | *Amphogona* | *Amphogona apsteini* |
| Anthoathecata | Corymorphidae | *Corymorpha* | *Corymorpha verrucosa* |
| | Porpitidae | *Porpita* | *Porpita porpita* |
| | Pandeidae | *Leuckartiara* | *Leuckartiara octonema* |
| | | *Amphinema* | *Amphinema rugosum* |
| | Proboscidactylidae | *Proboscidactyla* | *Proboscidactyla ornata* |
| Leptothecata | Aequoreidae | *Aequorea* | *Aequorea macrodactyla* |
| | | *Aequorea* | *Aequorea pensilis* |
| | | *Zygocanna* | *Zygocanna buitendijki* |
| | Eirenidae | *Eirene* | *Eirene hexanemalis* |
| **Cubozoa** | | | |
| Carybdeida | Tripedaliidae | *Tripedalia* | *Tripedalia maipoensis* |
| | Carukiidae | *Malo* | *Malo filipina* |
| | | *Morbakka* | |
| **Ctenophora** | | | |
| Beroida | Beroidae | *Beroe* | |
| | | *Beroe* | *Beroe forskalii* |
| Cydippida | Pleurobrachiidae | *Pleurobrachia* | *Pleurobrachia globosa* |
| Lobata | Bolinidae | *Bolinopsis* | *Bolinopsis rubripunctata* |
| | | *Bolinopsis* | |
| | Leucotheidae | *Leucothea* | |
| | Ocyropsidae | *Ocyropsis* | *Ocyropsis crystallina* |
| **Siphonophora** | | | |
| Siphonophorae | Agalmatidae | *Agalma* | *Agalma okenii* |
| | Forskaliidae | *Forskalia* | *Forskalia tholoides* |
| | Physaliidae | *Physalia* | *Physalia physalis* |
| | Rhizophysidae | *Rhizophysa* | |
| | Diphyidae | *Muggiaea* | *Muggiaea atlantica* |

as the lack of knowledge about local jellyfish occurrences and ecology, this citizen science dataset can meaningfully contribute to our understanding of Hong Kong's marine biodiversity. The HKJP has produced important information on the previously unrecorded presence of additional species of scyphozoans, hydromedusa, and even cubozoans, as well as documenting general occurrence patterns [12, 33, 39], showing the potential for citizen science to continue adding to knowledge of local jellyfish.

## DATA AVAILABILITY

The dataset described here is hosted on the GigaScience Press GBIF page [36].

## ABBREVIATIONS

GBIF: Global Biodiversity Information Facility; HKJP: Hong Kong Jellyfish Project; IPT: Integrated Publishing Toolkit.

## DECLARATIONS

### Ethics approval

Not applicable.

### Consent for publication

Not applicable.

### Competing interests

The author(s) declare that they have no competing interests.

### Authors' contributions

JT: conceptualization, data curation, formal analysis, funding acquisition, investigation, methodology, project administration, resources, software, validation, visualization, writing the original draft. YF: data curation, validation, writing, review, and editing. MJYL: data curation, validation, writing of the original draft. LJF: conceptualization, supervision, visualization, writing, review, and editing.

### Authors' information

JT is the founder and principal investigator of the Hong Kong Jellyfish Project. LJF has been an academic advisor throughout the Hong Kong Jellyfish Project. YF is a curator at GigaScience Press. MJYL is a member of the GBIF Asia regional support team.

### Funding

Not applicable.

### Acknowledgements

The authors would like to thank all the citizen scientists who contributed observations of jellyfish to the Hong Kong Jellyfish Project during 2021–2023.

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
