## [Reviewer Report]

Comments on revised manuscriptThe authors have addressed all my comments, and I confirm that a CC0 1.0 Universal Public Domain Dedication has now been ascribed to the GBIF dataset. In addition, the authors have updated the text in the main body of the manuscript and have corrected the erroneous entry of Macao as being in Hong Kong waters.  Furthermore, the authors have addressed the comments from the other reviewers. In particular, the inclusion of a Table with taxonomic scope of the Hong Kong Jellyfish Project jellyfish, in response to comments from Reviewers 2 and 3, is a very nice addition to the revised manuscript.  I consider these changes acceptable for publication.Comments on revised manuscriptThe authors have addressed all my comments, and I confirm that a CC0 1.0 Universal Public Domain Dedication has now been ascribed to the GBIF dataset. In addition, the authors have updated the text in the main body of the manuscript and have corrected the erroneous entry of Macao as being in Hong Kong waters.  Furthermore, the authors have addressed the comments from the other reviewers. In particular, the inclusion of a Table with taxonomic scope of the Hong Kong Jellyfish Project jellyfish, in response to comments from Reviewers 2 and 3, is a very nice addition to the revised manuscript.  I consider these changes acceptable for publication.

---

## [Editor Report]

Editor’s AssessmentThis Data Release present a dataset of jellyfish sightings collected by citizen scientists from 2021 through 2023 within Hong Kong waters. Citizen scientists submitted photographs and contextual metadata (including time, date, and location) via multiple sources and this dataset was harmonised and shared openly via the GBIF (Global Biodiversity Information Facility) repository. The end product presented here being three years of data and a total of 1,020 usable observations in GBIF. Sightings were validated using references from the literature, and after deposition and further validation of the data via the IPT tool, peer review helped fix a few remaining unreliable observations. These data should be important to address the research gap in jellyfish presence and distribution in Hong Kong by updating old records and adding new ones.Editor’s AssessmentThis Data Release present a dataset of jellyfish sightings collected by citizen scientists from 2021 through 2023 within Hong Kong waters. Citizen scientists submitted photographs and contextual metadata (including time, date, and location) via multiple sources and this dataset was harmonised and shared openly via the GBIF (Global Biodiversity Information Facility) repository. The end product presented here being three years of data and a total of 1,020 usable observations in GBIF. Sightings were validated using references from the literature, and after deposition and further validation of the data via the IPT tool, peer review helped fix a few remaining unreliable observations. These data should be important to address the research gap in jellyfish presence and distribution in Hong Kong by updating old records and adding new ones.

---

## [Reviewer Report]

Upload additional filesDRR-202402-01-R01/stage_files/DRR-202402-01/Review MS/gx-DR-1708376984 EBM.pdfReviewer name and names of any other individual's who aided in reviewer Ephrime MetilloDo you understand and agree to our policy of having open and named reviews, and having your review included with the published papers. (If no, please inform the editor that you cannot review this manuscript.)YesIs the language of sufficient quality?YesPlease add additional comments on language quality to clarify if needed
Are all data available and do they match the descriptions in the paper? NoAdditional CommentsWhere the results? What were the partial identification of jellyfish records, even at genus level? What were the synthesis made by authors with the data gathered?Are the data and metadata consistent with relevant minimum information or reporting standards? See GigaDB checklists for examples <a href="http://gigadb.org/site/guide" target="_blank">http://gigadb.org/site/guide</a>NoAdditional CommentsQuite unclear. Same issues are raised as above.Is the data acquisition clear, complete and methodologically sound?YesAdditional CommentsThe Methods section was fairly clear. Figure 2 text is blurred. Better include the URL of this sample pageIs there sufficient detail in the methods and data-processing steps to allow reproduction?YesAdditional CommentsFairly detailed and clear.Is there sufficient data validation and statistical analyses of data quality? NoAdditional CommentsBasis and justification of data (mainly photographs) rejections needs to be clearly described. Is the validation suitable for this type of data?YesAdditional CommentsIs there sufficient information for others to reuse this dataset or integrate it with other data?YesAdditional CommentsAny Additional Overall Comments to the AuthorThe manuscript could have mentioned the limitations and challenges of the approach or methods outlined in the manuscript.RecommendationMajor Revision

---

## [Reviewer Report]

Upload additional filesDRR-202402-01-R01/stage_files/DRR-202402-01/Review MS/gx-DR-1708376984-Round1.pdfReviewer name and names of any other individual's who aided in reviewer Colin J AnthonyDo you understand and agree to our policy of having open and named reviews, and having your review included with the published papers. (If no, please inform the editor that you cannot review this manuscript.)YesIs the language of sufficient quality?YesPlease add additional comments on language quality to clarify if needed
Are all data available and do they match the descriptions in the paper? YesAdditional CommentsAre the data and metadata consistent with relevant minimum information or reporting standards? See GigaDB checklists for examples <a href="http://gigadb.org/site/guide" target="_blank">http://gigadb.org/site/guide</a>YesAdditional CommentsIs the data acquisition clear, complete and methodologically sound?YesAdditional CommentsIs there sufficient detail in the methods and data-processing steps to allow reproduction?YesAdditional CommentsIs there sufficient data validation and statistical analyses of data quality? YesAdditional CommentsI don't really think you need to do this for this manuscript. However, I think it would be nice to have visualization representing the data. Perhaps histograms with taxonomy or maybe a table with more specific information would be nice. While I trust your data curation process, I do wish I could have some sort of preliminary data visualization.Is the validation suitable for this type of data?YesAdditional CommentsIs there sufficient information for others to reuse this dataset or integrate it with other data?NoAdditional CommentsThis was the one thing. I think it would be nice to be more explicit about how users can download and integrate this data into their standard workflows. Again, maybe a figure would be nice. Then, just be specific within the text about what you've done to make the data usable.Any Additional Overall Comments to the AuthorFirst off, I would like to apologize for my delayed review. I always get frustrated when my manuscripts get stuck in review, so please accept my apology. Generally, I like this style of paper and I think you do a nice job. I appreciate anything that promotes an open and accessible scientific community. The largest thing I was thinking throughout the paper was: "Is this different enough from iNaturalist to validate a separate database?" I am sure it is. Just be explicit with how.  I have attached a pdf with specific comments throughout the manuscript document. Do your best to respond to each comment. If you disagree with my suggestions, that is okay! I will not pressure you to incorporate my thoughts or suggested citations. I look forward to the eventual publication of this manuscript. RecommendationMinor Revision

---

## [Reviewer Report]

Reviewer name and names of any other individual's who aided in reviewer Chris ArmitDo you understand and agree to our policy of having open and named reviews, and having your review included with the published papers. (If no, please inform the editor that you cannot review this manuscript.)YesIs the language of sufficient quality?YesPlease add additional comments on language quality to clarify if needed
Are all data available and do they match the descriptions in the paper? NoAdditional CommentsI work for GigaScience Press and I was invited to perform a Data Audit on the following GBIF Dataset:  Terenzini J, Fan Y, Liu M J, Falkenberg L J (2024). Jellyfish in Hong Kong: a citizen science dataset. GigaScience Press. Occurrence dataset https://doi.org/10.15468/s4qwyk accessed via GBIF.org on 2024-03-04.  I can report that there are 1,021 occurrence and 90% with coordinates. Species are included in the GBIF record.  The Geographic Scope of this study is the “Waters around Hong Kong Special Administrative Region” and includes Macao, Shenzhen, and Dapeng.  However, in the GBIF record, it states that: “All jellyfish observations from the HKJP were made in Hong Kong waters.”  I would recommend that the authors either change this statement to include Macao, Shenzhen, and Dapeng, or alternatively remove the Macao, Shenzhen, and Dapeng occurrence data from the GBIF record.  As an additional point, the GBIF record has been ascribed a CC-BY-4.0 license. However, it is GigaByte policy that a CC0 1.0 Universal Public Domain Dedication should be ascribed to the GBIF dataset. I would encourage the authors to follow GigaByte policy.
Are the data and metadata consistent with relevant minimum information or reporting standards? See GigaDB checklists for examples <a href="http://gigadb.org/site/guide" target="_blank">http://gigadb.org/site/guide</a>YesAdditional CommentsThe occurrence data are archived in GBIF.Is the data acquisition clear, complete and methodologically sound?NoAdditional CommentsThe Geographic Scope of this study is the “Waters around Hong Kong Special Administrative Region” and includes Macao, Shenzhen, and Dapeng.  However, in the GBIF record, it states that: “All jellyfish observations from the HKJP were made in Hong Kong waters.”  I would recommend that the authors either change this statement to include Macao, Shenzhen, and Dapeng, or alternatively remove the Macao, Shenzhen, and Dapeng occurrence data from the GBIF record.
Is there sufficient detail in the methods and data-processing steps to allow reproduction?YesAdditional CommentsIs there sufficient data validation and statistical analyses of data quality? YesAdditional CommentsIs the validation suitable for this type of data?YesAdditional CommentsIs there sufficient information for others to reuse this dataset or integrate it with other data?YesAdditional CommentsAny Additional Overall Comments to the AuthorAs an additional point, the GBIF record has been ascribed a CC-BY-4.0 license. However, it is GigaByte policy that a CC0 1.0 Universal Public Domain Dedication should be ascribed to the GBIF dataset. I would encourage the authors to follow GigaByte policy.RecommendationMajor Revision